# Uncertainty Relation between Detection Probability and Energy Fluctuations

**DOI:** 10.3390/e23050595

**Published:** 2021-05-11

**Authors:** Felix Thiel, Itay Mualem, David Kessler, Eli Barkai

**Affiliations:** Department of Physics, Institute of Nanotechnology and Advanced Materials, Bar-Ilan University, Ramat-Gan 52900, Israel; thiel@postoeo.de (F.T.); itaym_123@walla.com (I.M.); kesslerda1@gmail.com (D.K.)

**Keywords:** quantum search, dark and bright states, Zeno subspaces, ergodicity

## Abstract

A classical random walker starting on a node of a finite graph will always reach any other node since the search is ergodic, namely it fully explores space, hence the arrival probability is unity. For quantum walks, destructive interference may induce effectively non-ergodic features in such search processes. Under repeated projective local measurements, made on a target state, the final detection of the system is not guaranteed since the Hilbert space is split into a bright subspace and an orthogonal dark one. Using this we find an uncertainty relation for the deviations of the detection probability from its classical counterpart, in terms of the energy fluctuations.

## 1. Introduction

A classical random walk on a finite graph explores the system completely. In a search process, a particle starts on some node of a graph and then searches for a target on another node. For classical walks on such structures, the arrival probability $P_{\det}$, the probability that the particle starting on one node is eventually detected at some other specified location, is unity. In that limited sense, a classical random walk search on a finite graph, is exhaustive, though time-wise it is inefficient. The exception with $P_{\det}\neq 1$ is obvious. If the graph is decomposable into several non-connected parts, the dynamics is not ergodic, and then exploration of at least part of the space is prohibited. Thus, the related recurrence problem becomes an issue only for an infinite system. For example, a classical random walk on an infinite lattice in dimension larger than two is non-recurrent [1,2,3].

A very different behavior is found for quantum walkers [4,5,6,7] on finite graphs that start localized on a node of the graph. The concept of quantum arrival is not well-defined, and instead we discuss the first detection; see Sec. II. Moreover, constructive interference may speed up search processes dramatically, under certain conditions [8]. However, destructive interference may divide the Hilbert space into two components called dark and bright, and this yields an inefficient search and an effect superficially similar to classical non-ergodicity, $P_{\det}<1$. More specifically, an observer performs repeated strong measurements, on a node other than the starting node, in an attempt to detect the particle [8,9,10,11,12,13,14,15,16,17,18,19,20,21,22]. In the time intervals between the measurements the dynamics is unitary. The rate of measurement attempts is 1/τ, where τ is a parameter of choice (see details below). Due to destructive interference, there might exist certain initial states $\psi_{\mathrm{in}}$ whose amplitude vanishes at the detected node at all times and this renders them non-detectable. Such initial conditions are called dark states and they are widely encountered [23]. In this case, the mean hitting time, i.e., the mean time for detection, is infinite [10]. This occurs even for simple models like quantum walks on a hyper-cube [8] or a ring [17]. More precisely, to get non-classical behavior for $P_{\det}$, the system has to have some symmetry property [10,24] (though in other cases, symmetry is responsible for the speedup of the quantum search). Generic initial states $\psi_{\mathrm{in}}$, are linear combinations of dark and bright states, the latter defined as states that are detected with probability one. It follows that a system starting in state $\psi_{\mathrm{in}}$ has a probability to be detected that lies somewhere between zero and unity. The challenge is to find $0\leq P_{\det}\leq 1$.

A formal solution for the detection probability was found in [8,24] and obtained explicitly for a few examples [25]. Since $P_{\det}$ is non-trivial, as compared with its classical counterpart, we will find its bound. This is derived via an uncertainty relation [24],
$$\Delta PVar{\left(H\right)}_{d}\geq {\left|d\left[H,D\right]\psi_{\mathrm{in}}\right|}^{2}.$$
Here $\Delta P=P_{\det}-|\langle \psi_{\mathrm{in}}d{\rangle |}^{2}$ is the deviation of the total detection probability from the initial probability of detection, $Var{\left(H\right)}_{d}$ is the variance of the energy in the detected state |*d*〉, and on the right hand side of the inequality we have the commutator of *H* and the measurement projector D=|d〉〈d|. Generally, the uncertainty principle [26], describes the deviation of the quantum world from classical Newtonian mechanics. Our goal is very different. We present an approach that shows how quantum walks, depart from the corresponding classical walks. The relation does not depend on the sampling time τ, and in fact is valid for a large class of measurement protocols. Recently, a time-energy uncertainty-like relation, was found in [27], with a focus on the time to detect a particle returning to its origin.

Note that the uncertainty principle discussed here, an exact formal solution to the problem, and an upper bound based on symmetry, were presented recently in a short communication [24]. Here, our goal is to provide a more detailed account of the problem. Below, we will show how to improve the lower bound obtained from the uncertainty principle. We also extend our previous study providing an upper bound based on an uncertainty principle, Equation (Equation 18) below. A key element of this work is to explore the basic ingredients of the uncertainty principle, for systems subject to repeated measurements. For large systems and depending on the initial condition, the lower bound in the above mentioned uncertainty relation, can be far from the exact result. However, improved bounds, which are based on related principles, can be very accurate. For example, we found a lower and upper bound on the detection problem of a hyper-cube in *d* dimensions, these coincide tightly with the exact result [24]. Thus, the focus here is on an elementary uncertainty principle for an evolution interspersed with measurements, with a form that relies on a commutation relation, in the spirit of the standard approach.

## 2. Model and Notation

We consider quantum dynamics on a finite graph. The evolution free-of-measurement is described by a time independent Hamiltonian *H* and the corresponding unitary propagator. Examples include a single particle on a graph, where *H* is a tight-binding Hamiltonian, e.g., the dynamics of a particle on a ring with hops to nearest neighbours. The theory is valid in generality, e.g., by identifying the graph with a Fock space, one can describe the dynamics of a many-body system, see an example in Ref. [27]. Initially, the particle is in state $|\psi_{\mathrm{in}}\rangle$, which could be a state localized on a node of the graph.

An extensively studied measurement protocol, exploits stroboscopic measurements at times τ,2τ,… in an attempt to detect the particle in state |d〉 (see [28] for the case of a general distribution of times between measurements). The detected state |d〉 can be a node of the graph or an energy eigenstate of *H*, etc. Specifically, the measurement, if successful, projects the wave packet onto state |d〉; otherwise this state is projected out and the wave function renormalised (see below). Between the measurement attempts the evolution is unitary and described by U=exp(−iHτ) and here ℏ=1. The outcome of a measurement is binary: either a failure to detect (no) or success (yes). Thus, the string of measurements yields a sequence no, no, and in the n-th attempt a yes, though the final success is not guaranteed. Once we detect the particle, we are done, and we say that nτ is the time it took to detect the particle in state |d〉. Fixing *H* and τ and repeating this measurement process does not mean that the particle is always detected. The question addressed here is what is the probability that the particle is detected at all, in principle after an infinite number of measurement attempts. This is the total detection probability $P_{\det}$.

This model is the quantum version of the first passage time problem [2,3]. However, the measurements back react and modify the dynamics. Specifically, each individual measurement is described with the collapse postulate. Namely, if the system’s wave function is |ψ〉 at the moment of the detection attempt, the amplitude of finding the particle at |d〉 is 〈d|ψ〉. As mentioned, if the particle is detected we are done. If not, the amplitude of the quantum state on the detector is set to zero, the wave function renormalised, and the evolution free of measurements continues until the next measurement, etc. Mathematically, the failed detection transforms |ψ〉, i.e., the wave function just before measurement, to N(1−D)|ψ〉 where *N* is the normalization factor, and D=|d〉〈d| is the measurement projector.

## 3. Lower Bound Using the Propagator

A bright state is an initial condition that is eventually detected with probability one and $P_{\det}\left(\psi_{\mathrm{in}}\right)=1$, while a dark state is never detected, and $P_{\det}\left(\psi_{\mathrm{in}}\right)=0$. Following [24,25], it is not difficult to show that if |β〉 is a normalised bright state then $U^{†}|\beta \rangle$ is bright as well. We will soon present simple arguments to explain this statement, but first let us point out its usefulness. Clearly it follows that ${\left(U^{†}\right)}^{k}|\beta \rangle$ is bright when *k* is a non-negative integer. As a bright-state seed consider the following state $|\beta \rangle =U^{†}|d\rangle$. |β〉 is an obvious bright state since it is detected in the first measurement attempt with probability one, because $\langle d|UU^{†}|d\rangle =1$. It therefore follows that the states
(1)$$U^{†}|d\rangle ,\cdots ,{\left(U^{†}\right)}^{k}|d\rangle \cdots$$
are all bright.

Why is $U^{†}|\beta \rangle$ bright in the first place? As is well known, the energy basis is complete. Less well known is that we may construct a complete set of stationary eigenstates of *H* which are either bright or dark [25]. In [25], we presented a formula for the stationary dark and bright states, making this statement more explicit, but for now all we need to know is that the finite Hilbert space can be divided into these two orthogonal subspaces denoted $H_{B}$ (bright) and $H_{D}$ (dark). Let $|E_{j}^{D}\rangle$ be a specific stationary dark state, and *j* is an index enumerating this family of states, and $E_{j}$ the corresponding energy. Then clearly $\langle E_{j}^{D}|U^{†}|\beta \rangle =0$ since the dark state is orthogonal to the bright one and $\langle E_{j}^{D}|U^{†}$ gives a phase $\exp(iE_{j}\tau )$. It follows that $U^{†}|\beta \rangle$ has no dark component in it and hence it is bright. Thus, the whole approach is based on the fact that we may divide a finite Hilbert space to a dark and bright subspaces (see also [29,30,31]).

We notice in the same way that the sequence
(2)$$U|\beta \rangle ,\cdots U^{k}|\beta \rangle \cdots$$
is also bright. As a consequence, since $U^{†}|d\rangle$ is bright, the state $UU^{†}|d\rangle$ is also bright. So when starting at the state $|\psi_{\mathrm{in}}\rangle =|d\rangle$, the system is detected with probability one, as was shown previously using other methods [12,28].

We now see that
(3)$$|d\rangle ,U^{†}|d\rangle \cdots$$
are bright states. However they are not orthogonal. We may choose the first two terms and construct two orthogonal bright vectors
(4)$$|{\tilde{\chi }}_{1}\rangle =|d\rangle \;and\;|{\tilde{\chi }}_{2}\rangle =N\left(|d\rangle -\frac{U^{†}|d\rangle }{|d\rangle U^{†}|d\rangle }\right)$$
where *N* is a normalization constant and |d〉, the detected state, is not an eigenstate of $U^{†}$. We note that using a similar approach it is not difficult to obtain further orthogonal bright states. Furthermore, the infinite sequence in Equation (Equation 3) is over-complete. To construct a bright space, we must find a set of orthogonal vectors forming a basis, using for example the Gram–Schmidt method. Formally the bright space is
(5)$$H_{B}=Span\left\{|d\rangle \;,U|d\rangle \;,U^{2}|d\rangle \;,\cdots \right\}$$
where Span{…} is the set containing all linear combinations of vectors of the states in the parentheses. Below we will construct this space explicitly for some simple examples, however in general this demands crunching linear algebra. For this reason the here-presented uncertainty relation, is useful in many cases.

When we have found a complete set of orthonormal basis vectors, which are either dark or bright states, the detection probability is given by [8];
(6)$$P_{\det}=\sum_{\tilde{\beta }\in H_{B}}|\langle \tilde{\beta }|\psi_{\mathrm{in}}{\rangle |}^{2},$$
where the summation is over a bright basis which as usual has many representations. It follows that
(7)$$P_{\det}\geq |\langle {\tilde{\chi }}_{1}|\psi_{\mathrm{in}}{\rangle |}^{2}+|\langle {\tilde{\chi }}_{2}|\psi_{\mathrm{in}}{\rangle |}^{2}.$$
We then find
(8)$$P_{\det}\geq \frac{|\langle d|U|\psi_{\mathrm{in}}\rangle {|}^{2}}{1-{\left|\langle d|U|d\rangle \right|}^{2}}$$
provided that the initial state and the detected one are orthogonal $\langle d|\psi_{\mathrm{in}}\rangle =\langle {\tilde{\chi }}_{1}|\psi_{\det}\rangle =0$. Note that $U|\psi_{\mathrm{in}}\rangle$ and U|d〉 are solutions of the Schrödinger equation in the absence of measurement starting with $|\psi_{\mathrm{in}}\rangle$ and |d〉, respectively, hence the lower bound relates the dynamics of a measurement-free process at time τ to the detection probability, which is the outcome of the repeated measurement process. The bound generally depends on τ and at least in principle one may search for τ that maximizes its right hand side. When τ is small, we may expand the propagator to second order $U∼1-iH\tau -H^{2}\tau^{2}/2$ and then find
(9)$$P_{\det}\geq \frac{|\langle d|H|\psi_{\mathrm{in}}\rangle {|}^{2}}{Var{\left(H\right)}_{d}}$$
where
(10)$$Var{\left(H\right)}_{d}=\langle d|H^{2}|d\rangle -\langle d|H|d\rangle^{2}$$
characterises the fluctuation of the energy in the detected state. Thus, the detection probability is bounded by the transition matrix from the initial to the detected state divided by the fluctuations of the energy in the latter. What comes to us as a surprise is that this result is valid for practically any τ, as we will show after a few remarks.

**Remark** **1.**
*Our results are valid for finite size systems, like finite graphs. For infinite systems, it is not always possible to divide the Hilbert space into two sub-spaces dark and bright. For example for a one-dimensional tight-binding quantum walk on a lattice, with non-zero jump amplitude to nearest neighbours only, starting on a node called the origin and measuring there, the non-zero detection probability is less than unity [17]. This means that |d〉 is not bright for an infinite system, which is physically obvious as the wave packet can spread to infinity and hence the particle can escape detection.*


**Remark** **2.**
*The states $|d\rangle ,\ldots U^{k}|d\rangle \ldots$ are bright and similarly with $U^{†}$. Can we find a bright state $|\beta^{\prime }\rangle$ orthogonal to these states? The answer is negative, and hence these states can be used to span the bright subspace. Assume $|\beta^{\prime }\rangle$ is an initial condition $|\psi_{\mathrm{in}}\rangle$ which is bright. At the first measurement, at time τ, the amplitude of detection is $\langle d|U|\beta^{\prime }\rangle$. However, this is zero by the assumption that $|\beta^{\prime }\rangle$ is orthogonal to the just obtained set of bright states. We may continue with this reasoning for the second, third, etc measurements, and we see that detection amplitude of $|\beta^{\prime }\rangle$ is always zero. It follows that state $|\beta^{\prime }\rangle$ is not bright.*


## 4. Uncertainty Relation

Let |β〉 be a bright state, then also f(H)|β〉 is bright where f(.) is an analytical function. Indeed similar to the previous section $\langle E_{j}^{D}|f\left(H\right)|\beta \rangle =0$ and hence the state f(H)|β〉 is orthogonal to the dark states, meaning it is bright. As we showed already the state |d〉 is bright so we find a sequence of bright states
(11)$$|d\rangle ,H|d\rangle ,\cdots ,H^{k}|d\rangle \cdots .$$
We use the same approach as in the previous section, namely we use the first two states and find two orthonormal bright states
(12)$$|\chi_{1}\rangle =|d\rangle \;and\;|\chi_{2}\rangle =N\left(|d\rangle -\frac{H|d\rangle }{\langle d|H|d\rangle }\right).$$
The normalization constant is given by ${\left|N\right|}^{2}={\left({\langle H\rangle }_{d}\right)}^{2}/Var{\left(H\right)}_{d}$ where ${\langle H\rangle }_{d}=\langle d|H|d\rangle$. Since *H* is Hermitian, inserting Equation (Equation 12) in Equation (Equation 7), and assuming no overlap of the initial state with the detection one, $\langle d|\psi_{\mathrm{in}}\rangle =0$ we get Equation (Equation 9). Thus, that formula is valid for any τ.

For the more general case when the initial overlap with the detection state is not zero, we define
(13)$$\Delta P=P_{\det}-|\langle d|\psi_{\mathrm{in}}{\rangle |}^{2}.$$
Since $\left|\langle d\right|\psi_{\mathrm{in}}{\rangle |}^{2}$ is the square of the overlap of the initial state and the detected one, it gives the probability to detect the particle in a single-shot measurement at time t=0. So ΔP is the difference between the probability of detection after repeated measurements and the initial probability of detection. Using Equations (Equation 7) and (Equation 12) and $\langle d|H\left(D-1\right)|\psi_{\mathrm{in}}\rangle =\langle d|\left[H,D\right]|\psi_{\mathrm{in}}\rangle$ we find
(14)$$\Delta PVar{\left(H\right)}_{d}\geq {\left|\langle d|\left[H,D\right]|\psi_{\mathrm{in}}\rangle \right|}^{2}.$$
Here $\left[H,D\right]$ is the commutator of the Hamiltonian and the projector describing the measurement. If $|\psi_{\mathrm{in}}\rangle =|d\rangle$ the right hand side is equal to zero and we find $P_{\det}\geq 1$, hence here the uncertainty relation indicates that the detection probability is unity.

**Remark** **3.**
*The matrix element 〈d|H|d〉 can always be set to be non-zero by a global shift of the energy, hence $|\chi_{2}\rangle$ is well defined. In the final result we can switch back to any choice of energy scale, since the $Var{\left(H\right)}_{d}$ is insensitive to the definition of the zero of energy.*


**Remark** **4.**
*For the stroboscopic sampling, recurrences and revivals imply that special sampling times τ defined through ΔEτ=2πk exhibit resonances [12,17] such that the bounds based on energy are invalid. Here ΔE is the energy difference between any pair of energy levels in the system. In this case, the starting point of the analysis should be Equation (Equation 1) and not Equation (Equation 11).*


**Remark** **5.**
*We believe, that our results are generally valid for other detection protocols, for examples when we sample the system randomly in time, following a Poissonian process [11,28]. Indeed the state $|\chi_{1}\rangle$ is bright, for generic measurement protocols, beyond the stroboscopic one [28]. Similarly, stationary dark states are generically dark. However, so far we have not proven that all the bright states of the stroboscopic protocol, are generically bright, under arbitrary repeated measurement protocols. In principle, an amplitude on a node of the system, can be zero at some set of times, and if one chooses these special and non-typical times for measurements, the detection probability can be set to zero (e.g., the measure zero exceptional points in the previous remark). A mathematical study of our main result, for general repeated measurement protocol, is left for future work.*


## 5. The Reverse Dark Approach

Assume |δ〉 is a normalised dark state, then as before the states $U^{k}|\delta \rangle ,H^{k}|\delta \rangle ,f\left(H\right)|\delta \rangle$ are dark as well. In fact since by definition a system initially in a dark state $|\psi_{\mathrm{in}}\rangle =|\delta \rangle$ is never detected, it follows immediately that $U^{k}|\delta \rangle$ and $H^{k}|\delta \rangle$ are dark (see remark below). However, there is no symmetry between dark and bright states, in the sense that, every system has at least one bright state |d〉, but not every system necessarily has a dark state. As we showed in [25], totally bright systems have non-degenerate energy levels and all the energy eigenstates have a finite overlap with the detected state. Still, let us assume that we find a state |δ〉, which is dark, we can then apply nearly the same strategy as before to find an upper bound for $P_{\det}$. We use |δ〉 and H|δ〉 to construct two orthonormal dark states
(15)$$|\xi_{1}\rangle =|\delta \rangle ,\;\;|\xi_{2}\rangle =N\left(|\delta \rangle -\frac{H|\delta \rangle }{\langle \delta |H|\delta \rangle }\right),$$
where *N* is a normalization constant. Clearly here we assume that the dark state |δ〉 is not a stationary state of the system, since otherwise $|\xi_{2}\rangle$ is not defined. Analogous to Equation (Equation 6), the detection probability is [25]
(16)$$P_{\det}=1-\sum_{\tilde{\delta }\in H_{D}}|\langle \tilde{\delta }|\psi_{\mathrm{in}}{\rangle |}^{2}.$$
Here the summation is over a basis of the subspace $H_{D}$. Since all the terms in the sum are clearly non-negative
$$P_{\det}\leq 1-|\langle \xi_{1}|\psi_{\mathrm{in}}{\rangle |}^{2}-|\langle \xi_{2}|\psi_{\mathrm{in}}{\rangle |}^{2}.$$
For simplicity, assume that $\langle \delta |\psi_{\mathrm{in}}\rangle =0$. Then using the normalization *N*, we find
(17)$$P_{\det}\leq 1-\frac{|\langle \delta |H|\psi_{\mathrm{in}}\rangle {|}^{2}}{Var{\left(H\right)}_{\delta }}.$$
Now we have an upper bound. As mentioned in the introduction, the detection probability even for small quantum systems on a graph, can be less than unity, unlike the corresponding classical walks. So this is a useful bound provided that we can identify |δ〉. Notice that the variance of energy is now obtained with respect to the dark state |δ〉. Of course, this state is dark with respect to a state |d〉 so while the detector does not appear explicitly in our formula, it is obviously important.

We now relax the condition $\langle \delta |\psi_{\mathrm{in}}\rangle =0$. Let $P_{ND}=1-|\langle \delta |\psi_{\mathrm{in}}{\rangle |}^{2}$ be the probability that initially the system is not in the dark state |δ〉 (hence the subscript ND). We consider the deviation $\delta P=P_{ND}-P_{\det}$, and find
(18)$$\delta PVar{\left(H\right)}_{\delta }\geq |\langle \delta |\left[H,|\delta \rangle \langle \delta |\right]|\psi_{\mathrm{in}}\rangle {|}^{2}.$$
Unlike the lower bound Equation (Equation 14), here the commutator on the right hand side, is between the Hamiltonian and a projector of the dark state, while previously the commutator was of *H* and the projector of the detector state.

## 6. Further Improvement of the Uncertainty Relation

When $\langle d|[H,D|\psi_{\mathrm{in}}\rangle =0$, the uncertainty relation Equation (Equation 14) does not provide useful information on ΔP. Such situations can be found for quantum walks on a graph and we encounter them frequently in the example section. This effect is easy to understand. Assume that initially the particle is localised on a node of a graph while it is detected on a another distant node, in particular $\langle d|\psi_{\mathrm{in}}\rangle =0$. Then if the hopping amplitudes are short ranged, the matrix element of *H* between the two distant nodes is zero and then the lower bound is not useful.

For such cases, we consider two other orthonormal bright states
(19)$$|\chi_{1}\rangle =|d\rangle \;\mathrm{and}\;|\chi_{2}\rangle =N\left(|d\rangle -\frac{H^{s}|d\rangle }{\langle d|H^{s}|d\rangle }\right)$$
where *s* is a positive integer. When s=1 we get the previously examined case. The normalization is ${\left|N\right|}^{2}={\left({\langle H^{s}\rangle }_{d}\right)}^{2}/\mathrm{Var}{\left(H^{s}\right)}_{d}$ where as before ${\langle H^{s}\rangle }_{d}=\langle d|H^{s}|d\rangle$ and $\mathrm{Var}\left(H^{s}\right)\rangle_{d}=\langle d|H^{2s}|d\rangle -\langle d|H^{s}|{d\rangle }^{2}$. Using Equation (Equation 7) we find
(20)$$\Delta P\mathrm{Var}{\left(H^{s}\right)}_{d}\geq |\langle d|\left[H^{s},D\right]|\psi_{\mathrm{in}}\rangle {|}^{2}.$$
This relation between ΔP, and the commutator of $H^{s}$ and the projector *D* is clearly *s* dependent. We will follow two approaches. In the first, we choose the smallest *s*, such that the right hand side of the uncertainty relation is not equal to zero. The second is to choose *s* in such a way to maximize the lower bound of $P_{\det}$. In the first approach and if *H* is described by an adjacency matrix, *s* has a simple physical meaning as it is the distance between the assumed localised initial state and the detector state, see further details below. The first approach is the quickest way to gain insight, while the second can be used to systematically improve the result.

## 7. Quantum Walks on Graphs

We consider a quantum walk of a single particle on a graph, modelled with a tight-binding Hamiltonian. In our examples *H* is described by an adjacency matrix. Thus, the particle can occupy nodes of a graph, the edges/connections describe hopping amplitudes. All these amplitudes are identical and on-site energies are set to zero. In the schematic figures of graphs under investigation, e.g., Figure 1, the circle with the light interior on a vertex describes the measured state |d〉. We will focus on initial conditions localised on another node (full circle) and also on the initial condition spread uniformly on the graph. Our goal is to find exact expressions for $P_{\det}$ using simple examples and compare the latter to the uncertainty relation.

### 7.1. Finite Line

We consider a quantum walk on a finite line with *L* nodes, focusing on the example of L=5. The localised basis is |*r*〉 with r=1,⋯L, so one may think of the end points as reflecting boundaries. This will soon be compared with a ring which has periodic boundaries. The Hamiltonian reads:(21)$$H=\gamma \left(\begin{array}{ccccc} 0 & 1 & 0 & 0 & 0\\ 1 & 0 & 1 & 0 & 0\\ 0 & 1 & 0 & 1 & 0\\ 0 & 0 & 1 & 0 & 1\\ 0 & 0 & 0 & 1 & 0 \end{array}\right).$$
The hopping energy γ is set to unity hereafter. Below we modify the location of the detector, and see its effect on the detection probability and the uncertainty principle. For schematics and summary of the results, see Figure 1.

*The transition to |d〉=|1〉.* Measurement is made on the left most node |d〉=|1〉 which from symmetry is the same as the choice |d〉=|5〉. We use |d〉,H|d〉⋯ to construct the bright subspace and besides obvious normalization we get
$$H_{B}=\mathrm{Span}\left\{|1\rangle ,|2\rangle ,|1\rangle +|3\rangle ,2|2\rangle +|4\rangle ,2|1\rangle +3|3\rangle +|5\rangle \right\}.$$
Since the dimension of the Hilbert space is five and since the above states are linearly independent, we have no dark subspace. Hence we are done: the detection probability of any initial condition $|\psi_{\mathrm{in}}\rangle$ is unity.

**Remark** **6.***For a quantum walk on a finite line, i.e., a lattice stretching from* |*1*〉 *to* |*L*〉 *with a time independent H, and hops to nearest neighbours only (not necessarily translation invariant as in our example), any initial state $|\psi_{\mathrm{in}}\rangle$ is detected with probability one if the measurement is performed on the end points |1〉 or |L〉. This conclusion is immediate since the action of $H^{k}$ on the detected state* |*1*〉 *gives L linearly independent vectors, which is the dimension of the Hilbert space.*

Let us check the uncertainty principle starting with $|\psi_{\mathrm{in}}\rangle =|2\rangle$. We have $\mathrm{Var}{\left(H\right)}_{1}=\langle 1|H^{2}|2\rangle -\langle 1|H|{1\rangle }^{2}=1$ and the transition amplitude 〈2|H|1〉=1 hence
(22)$$P_{\det}\left(2\to 1\right)\geq \frac{{\left|\langle 2|H|1\rangle \right|}^{2}}{\mathrm{Var}{\left(H\right)}_{1}}=1.$$
So, here the uncertainty principle gives the exact result, since clearly $P_{\det}\left(2\to 1\right)\leq 1$.

For the other starting points in the system, namely the transitions |r〉→|1〉, we use s=r−1 namely *s* is equal to the distance between the initial state and detected one. Our results are summarized in Figure 1 and read: $P_{\det}\left(2\to 1\right)\geq 1,P_{\det}\left(3\to 1\right)\geq 1,P_{\det}\left(4\to 1\right)\geq 1/5,P_{\det}\left(5\to 1\right)\geq 1/10$. We see that more distant initial states depart from the exact result $P_{\det}=1$. If $|\psi_{\mathrm{in}}\rangle =|d\rangle$ we mentioned already that the uncertainty relation with s=0 gives $P_{\det}\geq 1$. This means that the particle is detected with probability one, $P_{\det}\left(1\to 1\right)=1$.

To date, we had no overlap between initial and detected states. For the uniform initial state $|\psi_{\mathrm{in}}\rangle =\sum_{r=1}^{5}|r\rangle /\sqrt{5}$ we find $P_{\det}\geq 6/25$, where we used s=1 (as mentioned the exact detection probability is unity). We will later optimise the choice of *s* to see how one may improve the prediction of the uncertainty relation.

*Detector on |d〉=|3〉.* We now consider the detection on the middle point |3〉, a case that yields further symmetry in the problem, compared with the situations when the detector is on the other locations. The states |d〉,H|d〉 and $H^{2}|d\rangle$ are bright and are easy to evaluate $\left\{|3\rangle \;,|2\rangle +|4\rangle \;,|1\rangle +2|3\rangle +|5\rangle \right\}.$ It is then easy to construct the dark space, as we have only two dark states orthogonal to these bright states, so the dark basis is $\left\{|\delta_{1}\rangle ,|\delta_{2}\rangle \right\}$ with $|\delta_{1}\rangle =\left(|2\rangle -|4\rangle \right)/\sqrt{2}$ and $|\delta_{2}\rangle =\left(|1\rangle -|5\rangle \right)/\sqrt{2}$. It is easy to understand why these states are dark, as they interfere destructively on the detector on |3〉 in such a way that the amplitude of detecting the particle there is zero. We also have $H|\delta_{1}\rangle =|\delta_{2}\rangle$ and $H|\delta_{2}\rangle =|\delta_{1}\rangle$. Using the dark states and Equation (Equation 16), we find that $P_{\det}\left(r\to 3\right)=1/2$ for any initial state |*r*〉. The exception is the return problem |3〉→|3〉, which is detected with probability one.

Considering the transition |2〉→|3〉 we may use the two uncertainty relations, one with the detector state and the second with the dark state $\delta_{2}$ to get:(23)$$\frac{1}{2}\leq P_{\det}\left(2\to 3\right)\leq \frac{1}{2}.$$
So here the uncertainty relations give the exact result. The same holds for the transition 1→3 and the other transitions are clearly identical from symmetry. These results are summarized in Figure 1.

*Detector on |d〉=|2〉.* The bright subspace is spanned by $|d\rangle ,H|d\rangle ,\cdots H^{3}|d\rangle$ namely
$$H_{B}=\mathrm{Span}\left\{|2\rangle \;,|1\rangle +|3\rangle \;,2|2\rangle +|4\rangle \;,2|1\rangle +3|3\rangle +|5\rangle \right\}.$$
Here $H^{4}|d\rangle$ is also bright but is easily shown to be a linear combination of |d〉 and $H^{2}|d\rangle$. From these states it is easy construct an orthonormal bright basis
(24)$$\left\{|2\rangle \;,\left(|1\rangle +|3\rangle \right)/\sqrt{2}\;,|4\rangle \;,\left(2|5\rangle +|3\rangle -|1\rangle \right)/\sqrt{6}\right\}.$$
Hence the dimension of the bright subspace is four, and the dark subspace has one state in it. This vector is orthogonal to the bright basis, Equation (Equation 24), and hence it is easy to see that it is given by
(25)$$|\delta \rangle =\frac{|1\rangle -|3\rangle +|5\rangle }{\sqrt{3}}.$$
We have H|δ〉=0; hence this state is a stationary state with an eigenvalue equal to zero. It is easy to see why this is a dark state: its amplitude on the detector is zero for any time. Using Equation (Equation 16), $P_{\det}=1-|\langle \delta |\psi_{\mathrm{in}}{\rangle |}^{2}$ and we get the exact values of $P_{\det}\left(1\to 2\right)=P_{\det}\left(3\to 2\right)=P_{\det}\left(5\to 2\right)=2/3$ and $P_{\det}\left(4\to 2\right)=P_{\det}\left(2\to 2\right)=1$. Compared with the case when the detector was on the edge we get values for $P_{\det}$ which depart from the classical case of unit detection probability. In Figure 1. we compare these results with the uncertainty principle: $P_{\det}\left(1\to 2\right)\geq 1/2,P_{\det}\left(3\to 2\right)\geq 1/2,P_{\det}\left(4\to 2\right)\geq 1,P_{\det}\left(4\to 2\right)\geq 1/14$, where again we took *s* to be the distance between initial and detected state. For the uniform initial state we get $P_{\det}=14/15$ while the uncertainty principle gives $P_{\det}\geq 13/25$ when we choose s=1.

It is easy to extend these results for a segment of length L=2k+1, and k=1,2,⋯. We measure on |2〉 or |L−1〉 and then we have one and only one dark state. This state is an eigenstate $|\delta \rangle =\left(1,0,-1,0,1,0,-1,\cdots \right)/\sqrt{k+1}$ hence
(26)$$P_{\det}\left(r\to 2\right)=\left\{\begin{array}{cc} \frac{k}{k+1} & \;if\;r\;is\;odd\\ 1 & \;r\;is\;even \end{array}\right.$$
On the other hand, if *L* is even we have no dark subspace and the detection probability is unity. Thus, depending on whether *L* is even or odd, we may get a dark subspace or a completely bright situation.

**Remark** **7.***Note that in the example, the conclusion that for even L, we have only bright states, is valid for the specified location of the detector |d〉=|2〉. In general one can find for the segment model, with even L, initial conditions and detectors sites with sub-optimal detection probability $P_{\det}<1$. For our case under study, consider L=4, and as before |d〉=|2〉. We obtain the bright space, using the same method as before, and we find that it is spanned by:*$$H_{B}=Span\left\{|2\rangle \;,|1\rangle +|3\rangle \;,2|2\rangle +|4\rangle \;,2|1\rangle +3|3\rangle \right\}.$$*It is easy to see that from these vectors, we can construct four orthonormal states, and since L=4, we have a complete bright space and an empty dark sub-space. This is unlike the case L=5 where we had only four such bright states, implying a dark sub-space which is not empty. It is easy to extend the claim for larger segments with an even L, implying as we announced, that for |d〉=|2〉 and even L, $P_{\det}=1$. Delving deeper into this issue, the eigenvalues of H for even and odd L are non-degenerate. This means that the dark states, are found from eigenvectors of H, with zero over lap with the detected state. Consider, for example L=8. Then it is easy to show, with a program like Mathematica, that all the eight stationary states (eigen-vectors of H) have non-zero overlap with localized states, |1〉,|2〉,|4〉,|5〉,|7〉,|8〉, while some of the stationary states have zero overlap with* |3〉 *and* |6〉. *This indicates that if we chose the detectors on the latter sites, we will get sub-optimal detection (but not on |d〉=|2〉 considered in the example under study).*

### 7.2. Enumeration of Paths Approach

Let H=γA where *A* is the adjacency matrix of some graph. We set the energy scale γ=1 as it does not control the detection probability (like the sampling rate 1/τ). Thus, for the Hamiltonian under investigation, all bonds in the system are identical and on-site energies are zero. So $H_{ii}=0$ while $H_{ij}=1$ if the states are connected, zero otherwise. An example is the segment of size five just considered. As mentioned, the detection probability for the return problem |d〉→|d〉 is unity, hence below we consider only the transition problem from some localised initial state |*r*〉 to another orthogonal localised state |d〉. In this case, we can use a path counting approach to find a useful bound for $P_{\det}$.

We have
(27)$$H^{s}|d\rangle =\sum_{s\;paths}|j\rangle$$
where the sum is over all states |*j*〉 which are endpoints of paths starting on |d〉 and whose length is *s*. Clearly
(28)$$N_{r\to d}\left(s\right)=\langle r|H^{s}|d\rangle$$
is the number of paths starting on |*r*〉 and ending at |d〉 whose length is *s*. We then find from the uncertainty relation
(29)$$P_{\det}\left(r\to d\right)\geq \frac{{\left[N_{r\to d}\left(s\right)\right]}^{2}}{N_{d\to d}\left(2s\right)-{\left[N_{d\to d}\left(s\right)\right]}^{2}}.$$
The denominator is the variance of $H^{s}$ in the detector state. For example, when |r〉 is the nearest neighbour of |d〉 we choose, s=1 and get
(30)$$P_{\det}\left(r\to d\right)\geq \frac{1}{\#n.n}$$
where in the denominator we have the number of nearest neighbours. This is the reason why for the example of the line we got for nearest neighbours $P_{\det}\geq 1/2$ while for the two edge states, which have only one nearest neighbour, we got $P_{\det}\geq 1$. The bound depends on *s* and this can be used to our advantage. What is clear is that we must choose *s* to be larger or equal to the distance between the starting point to the measured one, otherwise the numerator in Equation (Equation 29) is equal to zero.

### 7.3. The Benzene-Like Ring

We now consider the tight-binding Hamiltonian for a particle on a ring with six sites [17]
(31)$$H=\gamma \sum_{i=0}^{5}\left[|i\rangle \langle i+1|+|i\rangle \langle i-1|\right],$$
see the top left panel in Figure 2 for schematics. Here we use periodic boundary conditions and hence |6〉=|0〉. Similar to the previous example, the energy scale γ is irrelevant for the determination of $P_{\det}$, hence we may set γ=1 and then *H* is the adjacency matrix of the benzene-like ring. The repeated measurements are made on a site we call |d〉=|0〉.

With the method of enumeration of paths we consider three transitions, 1→0, 2→0 and 3→0. Since the detector on the ring has two nearest neighbours, $P_{\det}\left(1\to 0\right)\geq 1/2$. For the transition 2→0 and 3→0 we will use s=2 and s=3, respectively. Notice that we have only one path of length two for the transition 2→0, while for the 3→0 transition we have two paths of length three. Elementary path counting gives $N_{0\to 0}\left(4\right)-{\left[N_{0\to 0}\left(2\right)\right]}^{2}=2$ for s=2 while $N_{0\to 0}\left(6\right)-{\left[N_{0\to 0}\left(3\right)\right]}^{2}=22$ for s=3, hence we get $P_{\det}\left(2\to 0\right)\geq 1/2$ and $P_{\det}\left(3\to 0\right)\geq 2/11$. These results are summarized in the upper right panel of Figure 2. We now analyse the problem exactly.

In Equation (Equation 11) we showed that if $|\psi_{\mathrm{in}}\rangle$ is any linear combination of the states |0〉, H|0〉, $H^{2}|0\rangle$ and $H^{3}|0\rangle$ then it is bright. Furthermore, $H^{4}|0\rangle$ is bright, however to construct a basis for the bright subspace $H_{B}$ we need only the just mentioned states. Using |d〉=|0〉, the bright subspace is given by
$$H_{B}=\mathrm{Span}\left\{|0\rangle ,\frac{|1\rangle +|5\rangle }{\sqrt{2}},\frac{2|0\rangle +|2\rangle +|4\rangle }{\sqrt{6}},\frac{3|1\rangle +2|3\rangle +3|5\rangle }{\sqrt{22}}\right\}.$$
These states are nearly intuitively bright, for example the terms in the second $\left(|1\rangle +|5\rangle \right)/\sqrt{2}$, interfere constructively on the detector situated at 0. On the other hand, the state $\left(|1\rangle -|5\rangle \right)/\sqrt{2}$ is dark from symmetry. The free evolution of this state gives zero amplitude on the detector. Acting with *H* on this state and normalising we see that $\left(|2\rangle -|4\rangle \right)/\sqrt{2}$ is also dark which is again obvious from symmetry.

With this information, we can solve the problem exactly. For example, consider a particle starting on
(32)$$|1\rangle =\frac{1}{\sqrt{2}}\left(\underset{\mathrm{bright}}{\underset{︸}{\frac{|1\rangle +|5\rangle }{\sqrt{2}}}}+\underset{\mathrm{dark}}{\underset{︸}{\frac{|1\rangle -|5\rangle }{\sqrt{2}}}}\right).$$
As mentioned the first term is bright and the second dark, hence $P_{\det}\left(1\to 0\right)=1/2$. Similarly $P_{\det}\left(2\to 0\right)=1/2$ and $P_{\det}\left(3\to 0\right)=1$. The transition 5→0 is identical to 1→0 and similarly for 4→0. Let us now see how the lower and upper bounds work for this case (L=6).

*The 1→0 transition.* As mentioned |d〉=|0〉 and using Equation (Equation 31) we find $\mathrm{Var}{\left(H\right)}_{0}=2$. For the initial condition $|\psi_{\mathrm{in}}\rangle =|1\rangle$ we have zero initial overlap with the detector. We find
(33)$$\frac{1}{2}=\frac{{\left|\langle 0|H|1\rangle \right|}^{2}}{\mathrm{Var}{\left(H\right)}_{0}}\leq P_{\det}\leq 1-\frac{{\left|\langle \delta |H|1\rangle \right|}^{2}}{\mathrm{Var}{\left(H\right)}_{\delta }}=\frac{1}{2}$$
where we used as a dark state $|\delta \rangle =\left(|2\rangle -|4\rangle \right)/\sqrt{2}$ and $\mathrm{Var}{\left(H\right)}_{\delta }=1$. In this case, the lower and upper bound coincide giving the exact result $P_{\det}=1/2$.

*The 2→0 transition.* Since the matrix element 〈0|H|2〉=0, we now use Equation (Equation 20) with s=2 since the distance between the initial state and measured one is 2. We find
(34)$$\frac{1}{2}=\frac{|\langle 2|H^{2}|{0\rangle |}^{2}}{\mathrm{Var}{\left(H^{2}\right)}_{0}}\leq P_{\det}\leq 1-\frac{{\langle \delta |H|2\rangle |}^{2}}{\mathrm{Var}{\left(H\right)}_{\delta }}=\frac{1}{2}.$$
$\mathrm{Var}{\left(H^{2}\right)}_{0}=2,\mathrm{Var}{\left(H\right)}_{\delta }=1$ and $|\delta \rangle =\left(|1\rangle -|5\rangle \right)/\sqrt{2}$. Again the lower and upper bound coincide giving the exact result.

*The 3→0 transition*. We now set s=3 finding $\mathrm{Var}{\left(H^{3}\right)}_{0}=22$, while $|\langle 0|H^{3}|{3\rangle |}^{2}=4$ and this gives the mentioned lower bound $2/11\leq P_{\det}\leq 1$ which is to be compared with the exact result $P_{\det}=1$.

In general one could imagine several ways of how to improve the lower bound. The obvious one is to go beyond the calculation of the bright states |d〉 and $H^{s}|d\rangle$, namely consider a third bright state, but that means more algebra. Another option is to choose $H^{s_{1}}|d\rangle$ and $H^{s_{2}}|d\rangle$ as the starting point (we took $s_{1}=0$ and $s_{2}=1,2,3$). In the next section, we will improve the lower bound using a simple example. An upper bound can be tackled with symmetry arguments, which provide a different perspective [24].

### 7.4. Optimisations of the Lower Bound

Here we use a simple model of a system with a dangling bond, and consider the optimisation of the lower bound, i.e., we will soon search for the best choice of *s*. For the quantum walk on the line, when the measurement is made on the end point, we showed that the detection probability is unity. We add a perturbation to the system: one link perpendicular to the line. We will treat the example of a system with 6 nodes on the backbone and one dangling bond; see schematics Figure 3 and details in the Appendix A. Adding one dangling bond breaks the symmetry, but only partially in the sense that we still find that the detection probability is not generically unity. We note that strongly disordered systems with no symmetry exhibit a classical behaviour, namely $P_{\det}=1$ [24].

The notation used, the exact results, and the uncertainty principle are presented here in Figure 3, where the transition from a localised initial state to the detector on |0〉 is considered. The distance between the initial condition and the detected state is *s*. Now we turn to a simple optimisation of the bound, both for a localised initial state and for a uniform initial condition.

Consider the initial condition which is uniform $|\psi_{\mathrm{in}}\rangle =\sum_{r=0}^{6}|r\rangle /\sqrt{7}$. In this example we have only one dark state $|\delta \rangle =\left(|6\rangle -|3\rangle +|5\rangle \right)/\sqrt{3}$, and with this we find the detection probability $P_{\det}=20/21$. The uncertainty principle, with s=1 reads: (35)$$P_{\det}-\frac{1}{7}\geq \frac{|\langle 0|H\left(1-|0\rangle \langle 0|\right)|\psi_{\mathrm{in}}\rangle {|}^{2}}{\mathrm{Var}{\left(H\right)}_{0}}=\frac{1}{7}.$$
Here, 1/7 on the left hand side is the initial overlap, and we have chosen the detected node |d〉=|0〉. This gives $P_{\det}\geq 2/7$. To improve this bound, we made similar calculations with s=2,…,5; the results are presented in Table 1. The best lower bound is found for s=5, $P_{\det}\geq 167/253≃0.6117$. To further improve the bound, we depart from the pen and paper approach and instead we use a simple computer program. The results for both the uniform initial condition and a localised initial state are presented in Figure 4. This figure shows that increasing *s* yields a better result for the lower bound, but eventually the calculation saturates while exhibiting odd/even oscillation, Note that generally the lower bound calculations involve only the multiplication of matrices, while obtaining an orthonormal basis for the dark $H_{D}$ or bright $H_{B}$ subspaces, needed for the calculation of the exact expression for $P_{\det}$ Equations (Equation 6) and (Equation 16), demands considerably more work.

### 7.5. Other Examples

For not too large systems, it is relatively easy to write a simple program generating the bright states |d〉,H|d〉⋯ and then to perform the Gram–Schmidt procedure. This way we find an orthonormal basis for the bright space $H_{B}$. Furthermore, then we can find the detection probability using Equation (Equation 6). In Figure 2 we present some graphs and the corresponding detection probability. Here, as before, the starting point is a node of a graph, and the empty circle is the position of the detector. It is striking, that when the detector is placed on a symmetry centre of the graph, $P_{\det}$ may exhibit a deficit from the classical limit of unity. Since the classical detection probability on similar graphs is unity, the deviations are of interest. So as a rule of thumb we use systems with symmetry to find deviations from classical expectations.

All the examples presented here are based on rather simple graphs. This allowed us to obtain exact solutions rather easily and then estimate the performance of the uncertainty principle. For large and generally complex systems one needs a set of tools, and one cannot rely only on the method presented here. If the system is disordered, such that the energy spectrum is non-degenerate, and each stationary state has some finite overlap with the detected state, the detection probability is unity. In this case, the system behaves classically and there is no need at all for a lower bound. For systems with symmetry, and hence a degenerate spectrum, an upper bound based on symmetry was found. This relates the number of equivalent states in the system ν, i.e., states that are equivalent to the initial state with respect to the measured one, showing that $P_{\det}\leq 1/\nu$. Finally, here we used as a seed the bright state |d〉. In [24], we proposed a shell method, that searches for a bright state, with a strong overlap with the detected one. This allowed us to treat several large systems, like systems with loops, the hyper-cube, and also go beyond the examples with adjacency Hamiltonians treated here [24]. Further investigation of other structures like large glued trees [32,33], and fractals [34], is left for future work.

## 8. Discussion

This work relied on the partition of the Hilbert space into dark and bright sub-spaces. Probably the best known example, is the case of rapid measurements, τ→0 [35]. Then, when measuring on a node of the graph, it is not difficult to find a basis for this pair of sub-spaces. The measured node is obviously the bright sub-space, since the particle is detected with probability one at the first measurement. Localized initial states, on all the other nodes are dark, since the square modulus of the amplitude at the detected site is of order $\tau^{2}$ (unlike classical walks where the corresponding probability increases like τ) and hence $P_{\det}=0$ in this limit. It was later realised that the splitting of the Hilbert space into two components, does not need to rely on fast measurements [10,29,30,31]. The general mechanism behind this effect is destructive interference. Recently, we expressed the dark and bright sub-space of a general Hamiltonian in terms of its eigenstates [25]. Using this, we could find here a bound for the detection probability, in the form of an uncertainty principle.

The splitting of the Hilbert space into two components, resembles the splitting of a classical system into two disjoint components, namely ergodicity breaking. Let us assume we have such a classical system, which is split into non-connected domains *A* and *B*. We add a detector within one sector, say *A*. If we detect the particle, we know that it started in *A*, otherwise it started in *B*. In the quantum world we may start in a superposition state, with components in both the dark and bright sub-spaces. Hence, this situation is very different as compared with the classical case of ergodicity breaking, leading to non-trivial $P_{\det}$ even if *H* itself describes a fully connected system.

The uncertainty relation Equation (Equation 14) does not depend on the measurement frequency 1/τ and in that sense it is universal. However, in some cases, its right hand side is equal to zero. In particular, as shown here, this occurs when an initial localised state and a detected localised state are far from each other and the Hamiltonian describes a finite range of jump amplitudes. A second relation, Equation (Equation 20), depends on the free parameter *s*, allowing to connect between distant states, and this permits an easy calculation of a nontrivial lower bound for $P_{\det}$. We showed how to optimise the choice of *s*, improving the lower bound. As mentioned, more advanced methods, intended for larger systems, are discussed in [24]. A general strategy, to improve the results obtained here, is based on further collecting bright or dark states, instead of the two we used in Equations (Equation 7), (Equation 12) and (Equation 19). In principle, and in particular with a simple computer program, one may use more states with a Gram–Schmidt method, and gain tighter bounds than those found here. However, that comes at a cost, namely the theory becomes in some sense cumbersome, as compared to the uncertainty principle discussed here.

In this article we considered repeated strong measurement as the protocol of choice. Due to the wide range of quantum measurement theories, one must wonder how general are the results presented here? While the answer to this question is left for future work, we may speculate the following. The mechanism leading to dark states is in principle simple: the amplitude of the wave function at |d〉 is equal to zero forever. Hence any choice of a measurement theory or any measurement protocol, that is reasonably physical in the sense that it postulates that we cannot detect nor influence the state of the particle if the amplitude of finding it is zero, will yield the same dark states as for strong measurements. Still, we cannot claim any results for weak measurements [36]. We believe that our results, hold also for the well-known non-Hermitian approach, where the detection is modelled with a sink. For the limit of small τ it was shown [14,20] that one may use a non-Hermitian approach to model the strong detection protocol considered here. Hence, the two approaches have many things in common. Instead of stroboscopic sampling one may use temporal random sampling, for example sampling times drawn from a Poisson process [11,28]. Again we believe that this will not alter our results since the destructive interference is found also in this case. The fact that our results are τ independent is another argument for the generality of the approach.

The uncertainty principle investigated here, is different from the standard approaches [37,38,39,40]. These are roughly divided into two schools of thoughts. The text-book momentum-position uncertainty relation, is a measure of uncertainty in the state function. To verify it, one needs to perform two sets of measurements obtaining the uncertainty in *x* and *p* independently. The second, is the disturbance approach originating from the γ-ray thought experiment [37]. This dichotomy continued to attracted considerable research until recently [41,42,43,44,45,46,47,48]. Our approach is different from both and this is obviously related to the fact that we consider repeated measurements which back-react and modify the unitary evolution and also to the observable of interest: the detection probability. The uncertainty relation found here, can be extended to other observables. In [27], a time-energy relation was discovered for the fluctuations of the return time, with an interesting dependence on the winding number of the problem.

As for possible experimental observation, the quantum walk has already been demonstrated, using single neutral particles and site-resolved microscopy [49,50]. Usually, the focus is on the measurement of the propagation of the packet of particles. This demands what we may call a global measurement searching for the position of the particle at time *t*, while we are considering a spatially local measurement which detects the particle on a single target node of a graph. Such experiments, on the recurrence problem, were conducted in [51] with coherent light using strong projections, the number of repeated measurements was roughly forty. Thus, measurements of the quantum detection probability $P_{\det}$, and the uncertainty principle, are within reach.

Usual uncertainty relations are statements showing the departure of quantum reality from classical Newtonian mechanics. Here however, we are dealing with the departure of quantum search from its classical random walk counterpart. In Equation (Equation 14) we use the fluctuations of energy in the detected state $\mathrm{Var}{\left(H\right)}_{d}$ and it is natural to consider its meaning in a measurement protocol. The observer repeatedly attempts to detect the particle and once successful, namely the particle is detected, the particle is in state |d〉. Once the particle is detected we stop the monitoring measurements on |d〉. This means that in this second stage of the experiment the energy is a constant of motion. We now measure *H*. Hence repeating the protocol many times we have from the first stage of the experiment an estimate for $P_{\det}$ and from the second the variance of *H* in the detected state is obtained. It follows that at least in principle there is a physical meaning to the variance of *H* in the detected state, as these are the fluctuations after the particle is finally detected (if the particle is not detected, we do not record the energy). It follows that we may rewrite Equation (Equation 14) in a form that emphasizes the role of the state function. Since the final wave function, after a *successful* detection is $|\psi_{\mathrm{fin}}\rangle =|d\rangle$ we have
(36)$$\Delta P\mathrm{Var}{\left(H\right)}_{\psi_{\mathrm{fin}}}\geq {\left|\langle \psi_{\mathrm{fin}}|\left[H,D\right]|\psi_{\mathrm{in}}\rangle \right|}^{2}.$$
The same holds more generally for s≠1. Note that $\mathrm{Var}{\left(H\right)}_{\psi_{\mathrm{fin}}}$ is a constant of motion after the successful detection, since as mentioned we stop the repeated detection attempts once obtaining the yes click. This means that the observer does not need to measure the fluctuations immediately after the successful detection and there is no issue with the violation of the energy-time principle.

## Figures and Tables

**Figure 1 entropy-23-00595-f001:**
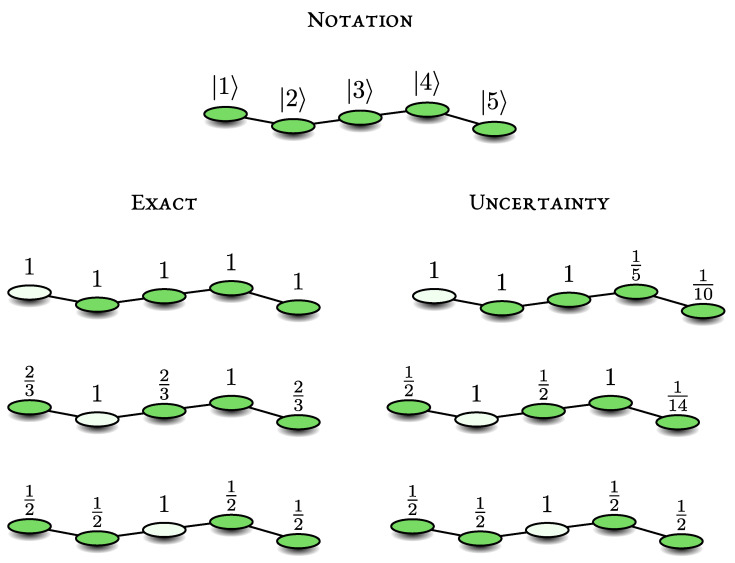
A quantum walk on a line with five sites. The initial condition is any localised state denoted with circles with an interior coloured green. The detected state |d〉 is the circle with a white interior. Here we present the notation used in the text, the exact results of $P_{\det}$ and the lower bound found using the uncertainty relation. For example, starting on |1〉 and measuring on |2〉 we have $P_{\det}=2/3$ while the uncertainty relation gives $P_{\det}\geq 1/2$.

**Figure 2 entropy-23-00595-f002:**
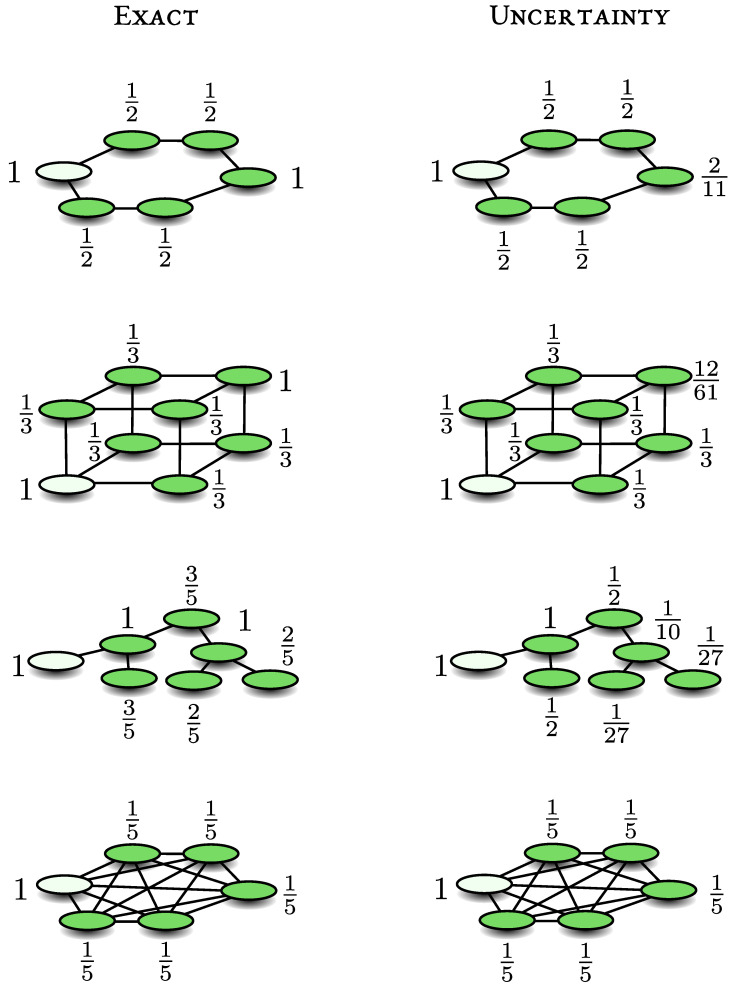
For a quantum walk starting on a node of the graph and measured elsewhere, we present the exact result for the detection probability, which depend’s on the location of the starting point. Shown also is the lower bound, obtained with the uncertainty principle. We choose *s* as the smallest integer, for which the uncertainty principle is non-trivial, namely the case where $\langle \psi_{\mathrm{in}}|H^{s}|d\rangle \neq 0$, this is the distance between the measured site and the initial condition.

**Figure 3 entropy-23-00595-f003:**
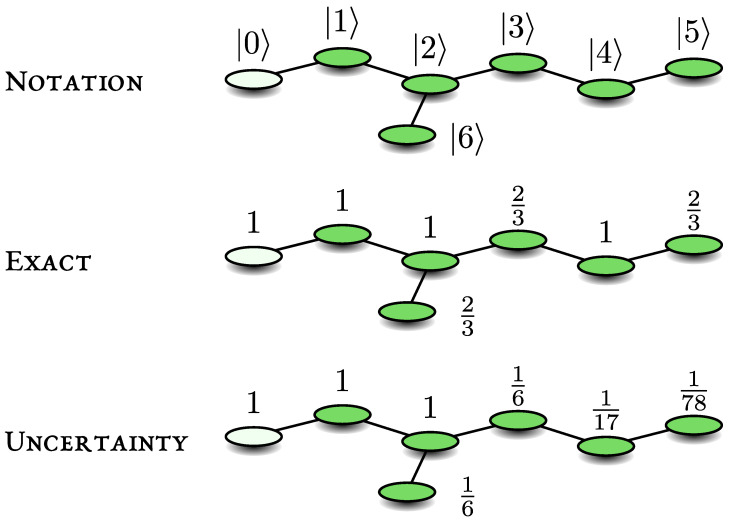
A system with a dangling bond, where the edge state |0〉 is repeatedly measured. Note that in the absence of the bond, i.e., when node |6〉 is removed, the detection probability is unity, no matter what is the starting point. To obtain the lower bound, using uncertainty, we choose *s* to be the shortest distance between the initial condition and the detected state. In Figure 4 we improve the bound for the transition |5〉→|0〉, which here gives $P_{\det}\geq 1/78$.

**Figure 4 entropy-23-00595-f004:**
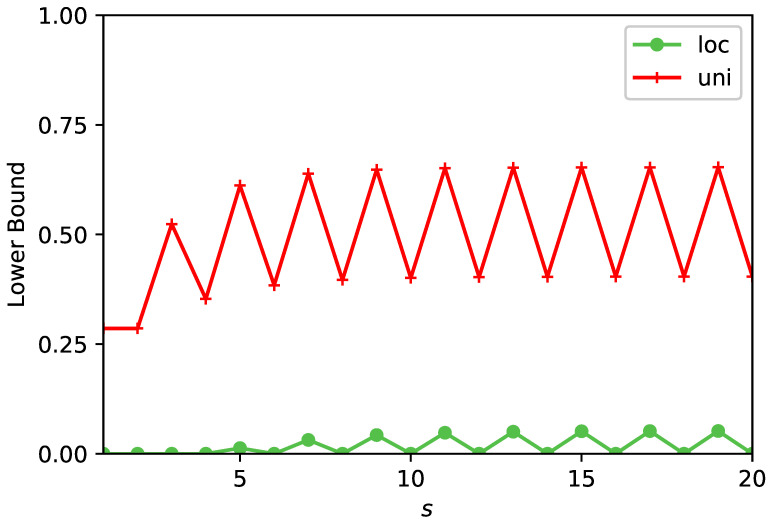
We demonstrate the optimisation of the uncertainty principle, for a system with a dangling bond. The Figure shows the lower bound for $P_{\det}$ versus *s*. The measurement is on node |0〉 and we consider two initial conditions: one localised $|\psi_{\mathrm{in}}\rangle =|5\rangle$ and the other uniform. See upper part of Figure 3 for schematics and notation. Starting on node |5〉 and when s≤4 the bound is equal to zero since if the initial and detected states are localised, *s* must be larger than the distance between these states to make the approach useful. Increasing *s* clearly leads to improvements of the bound. The exact results are $P_{\det}=20/21$ for the uniform initial condition, and $P_{\det}=2/3$ for the initially localised state.

**Table 1 entropy-23-00595-t001:** For the system with a dangling bond, the detection being on |0〉, the Table gives the lower bound for the uniform initial condition versus *s*. Increasing *s* in this range is improving the bound.

*s*	1	2	3	4	5
$P_{\det}\geq$	$\frac{2}{7}$	$\frac{2}{7}$	$\frac{11}{21}$	$\frac{42}{119}$	$\frac{167}{253}$

## Data Availability

Not relevant.

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
