# Peer review of "Uncertainty Relation between Detection Probability and Energy Fluctuations"

_entropy, 2021, doi:10.3390/e23050595_

Round 1

Reviewer 1 Report

I believe that this manuscript will be of interest to the readers of Entropy and can be published with some minor corrections. It follows rather closely a prior publication by the same authors (disclosed here as Ref. 24), with many similarities, however, this presentation is more focused and detailed on laying out examples of the process and details of the calculations that was left out that rather brief earlier paper.

Overall, the effect described here, namely the possibility of decomposing Hilbert space into orthogonal bright and dark sectors, referring to the ability for a target site to be reachable or not in a finite network. The eigenbases can be constructed recursively from that target site basis vector in linear combinations of the site basis using the Hamiltonian. So, for certain initial local or extended initial conditions (ie, those in the dark eigenbasis), that site might not be reached at all, or only with finite probability when the initial state is a combination from both sectors. As these constructions, here executed exactly for some small examples, become exceedingly tedious, lower and upper bounds are constructed. These bounds are also tested on those examples.

The discussion is straightforward and insightful, despite a good number of typographical errors and convoluted or repetitive statements. Overall, it is quite pedagogical. However, there are some issues I did not quite understand. First, the formalism is shown using the evolution operator and a big point is made out of the need for stroboscopic measurements. Yet, then it is argued that the rate of that measurement is actually irrelevant. Moreover, the formalism is shown to work for H itself as generating operator, or any function of H (such as U). So, that begs the question on what the need for any specific measurements is here. Would it be enough to say (like in the classical case) that the first passage PDF (if measured during an infinitesimal interval at time t) falls with time like some power of time and those integrated lead to P_det? (I assume from this that all results also apply to a discrete-time QW.)

It is commented on the role of symmetry but most larger graphs, of course, have little symmetry. So, it seems that, except for some important cases such as lattices, for most systems the dark sector might become rather irrelevant. Also, it is not clear from those small examples how the asymptotic performance of the bounds (ie, their tightness) under those conditions evolves, of course. Finally, I was wondering what one would find for a fractal network, where spectral have certain degeneracies?

Author Response

 We thank the referee for his/her generally supportive comments, and the constructive input. 

The referee raises the issue on our reliance on the stroboscopic protocol and then writes:

>Would it be enough to say (like in the classical case) that the first passage PDF >(if measured during an infinitesimal interval at time t) falls with time like some >power of time and those integrated lead to P_det? 

 We have several comments on this issue. First, we stated clearly (and as the referee mentioned in his report) that the results are general. For example, we expect them to hold for a Poisson sampling protocol. 

 However, if one fixes the measurement times, to some specific sequence, t_1, t_2, ... etc, one may encounter certain behaviours very different from classical random walks. It may happen, in quantum dynamics, that at some specified times, the amplitude on the detected state vanishes, this is related to periodicities and revival of the wave functions (we discuss this in  remark 4 and 5, briefly). If the measurement times t_1, t_2 coincide with these special times, we will never detect the particle.  This makes a general proof hard.  Even more technically,   assume that these times are generated from some multi-dimensional  distribution. We then need to average the results over an unspecified distribution. This step was performed only recently, assuming that the time intervals between measurements are IID random variables (our work from 2021). The math in this case is non-trivial. And we did not search for  an uncertainty principle in this case. Technically, the issue is the following. It is easy to show that the dark states are always dark, since they have an amplitude zero on the detected site. However, to prove that the bright states are detected with probability one, for any realisation of the measurement sequence (specific sequence) or for nearly all measurements sequence (besides measure zero) is not something we considered so far. 

The referee adds:

>It is commented on the role of symmetry but most larger graphs, of course, >have little symmetry. So, it seems that, except for some important cases such >as lattices, for most systems the dark sector might become rather irrelevant. 

In the context of quantum information, the search problem in large systems, is in many cases devoted to systems with symmetry built in them. This includes, for example glued trees (popular in computer science) and hyper-cubes in large d dimensions (mentioned in the text).  One reason to do so, is that the quantum search, to become efficient, actually exploits symmetry to build constructive interference, and speed up dramatically, the search (see ref. 8, work of Todd Brun).  Thus symmetry is a double sword. In some cases, i.e. some initial conditions, it can speed up search through constructive interference, and some times, it gives destructive  interference, and hence inefficient search. In some, sense this is one of the challenges of the field. Our work, quantifies the deviations of P_det from unity, but as we mention now in the introduction, it is not correct to think that quantum search on symmetric objects is non-useful. To put differently, if we break the symmetry (add disorder) we may encounter Anderson localization, and that is certainly bad for fast search. 

Finally, the referee points out correctly, that our examples focuses on  small systems, and wonders how this can be improved. 

In the new version of the manuscript, we discuss this issue briefly. One may, with a simple computer program, and some number crunching, find tighter bounds. However, one then loses the elegance of the theory. By that we refer to a simple and in our opinion elegant uncertainty principle which depends on a commutation relation, and energy fluctuations, maybe similar (but distinct) if compared with the usual uncertainty relation, from basic QM. We have however exposed the route to tighter bounds, instead of using a pair of bright states to find a lower bound on P_det,  see Eq. 7, one can use say k states (and we showed how to get them, in principle). In practice, this will be done on a computer, since one needs to find orthogonal states, which demands vector crunching, which of course is not a difficult task. We conclude here, that our manuscript is useful for any reader who wishes to find tighter bounds, with some additional work, since it exposed the basics of the problem. 

 Following the remarks made by the referee, we commented on these new points, in the introduction, remark 4 and 5, and further  Sec. titled other examples, and summary. We have also added ref.  for  glued trees and fractal structures, the latter mentioned in the referee report, since this will surely give further insight. 

Reviewer 2 Report

see comments to author 

Author Response

Thanks for the generally supportive review and the constructive remarks. Note that some of your comments, were addressed already in the reply to the first referee. In particular with respect to applicability and the scale up of the results to larger, systems. We will not repeat all our arguments here. 

>All of the figures are far removed from the sections where they are discussed. It >would save a reader some time if the figures and the text the reference them >are closer together or side-by-side.

Thanks. We submitted a Revtex file to the journal, which produced figures in proper places, for example this is evident in our arXiv version of the paper. Of course, we will solve this problem, with the editors.

> I am having a hard time seeing the practicality of the bound approach.

See reply to first referee. If interested, see how we generalise and extend the current approach, to find a tight bound for a hyper cube in (any) d dimension Ref. 24 in the paper. 

More specifically, the principles, outlined in this paper, can be used to improve the bound, while here we focused on an elementary formula, the uncertainty principle. As mentioned, we now discuss these issues further  in the MS.

> The referee wonders with respect to the novel aspects of the results compared to a  previous publication. 

Note that we have two upper bounds (as noticed by the referee).  One of them based on symmetry was presented previously. The other one using two dark states, is presented here for the first time. See. Eq. 18. Similarly, the bounds based on U, in the first part of the paper, are new.  The discussion on the meaning of the principle, in summary Eq. 36 is also new. We explain there what is the meaning of energy fluctuations in state |d>. This is physically not trivial in the sense that along the measurement process energy is not conserved.  Further the examples and discussion, serve a purpose that we think is worthy, in particular for an invited paper in a special issue. Further, previous ms was a short letter-like ms, which included many results, not focused on the uncertainty alone.

We added punctuation, and improved the presentation. 

We changed order of examples, as suggested by the referee.

Specific comments:

We now describe classical walks as exhaustive. 

We fixed grammatical problems raised by the referee.  Other comments and suggestions by the referee are included in the new version of the manuscript, the comments were very useful, thanks.  

 The referee, suggested that we consider the removing of one paragraph, that dealt with possible experiments. We opted to leave this paragraph, since we hope that our work will inspire real life applications.  By the way, these consider small systems, in a practical sense large systems in 2021 are a fantasy of theoreticians. Though, of course, the point of view of the referees, that large complex systems are of interest, is absolutely correct.